# Effects of Vitamin B12 Supplementation on Cognitive Function, Depressive Symptoms, and Fatigue: A Systematic Review, Meta-Analysis, and Meta-Regression

**DOI:** 10.3390/nu13030923

**Published:** 2021-03-12

**Authors:** Stefan Markun, Isaac Gravestock, Levy Jäger, Thomas Rosemann, Giuseppe Pichierri, Jakob M. Burgstaller

**Affiliations:** 1Institute of Primary Care, University and University Hospital Zurich, 8091 Zurich, Switzerland; stefan.markun@usz.ch (S.M.); levy.jaeger@usz.ch (L.J.); thomas.rosemann@usz.ch (T.R.); giuseppe.pichierri@usz.ch (G.P.); 2Horten Centre for Patient Oriented Research and Knowledge Transfer, University of Zurich, 8091 Zurich, Switzerland; isaac.gravestock@gmail.com

**Keywords:** vitamin B12, cognitive function, depressive symptoms, fatigue, RCT, meta-analysis

## Abstract

Vitamin B12 is often used to improve cognitive function, depressive symptoms, and fatigue. In most cases, such complaints are not associated with overt vitamin B12 deficiency or advanced neurological disorders and the effectiveness of vitamin B12 supplementation in such cases is uncertain. The aim of this systematic review and meta-analysis of randomized controlled trials (RCTs) is to assess the effects of vitamin B12 alone (B12 alone), in addition to vitamin B12 and folic acid with or without vitamin B6 (B complex) on cognitive function, depressive symptoms, and idiopathic fatigue in patients without advanced neurological disorders or overt vitamin B12 deficiency. Medline, Embase, PsycInfo, Cochrane Library, and Scopus were searched. A total of 16 RCTs with 6276 participants were included. Regarding cognitive function outcomes, we found no evidence for an effect of B12 alone or B complex supplementation on any subdomain of cognitive function outcomes. Further, meta-regression showed no significant associations of treatment effects with any of the potential predictors. We also found no overall effect of vitamin supplementation on measures of depression. Further, only one study reported effects on idiopathic fatigue, and therefore, no analysis was possible. Vitamin B12 supplementation is likely ineffective for improving cognitive function and depressive symptoms in patients without advanced neurological disorders.

## 1. Introduction

Declines in cognitive function, depressive symptoms, and idiopathic fatigue (fatigue of unknown cause) are highly prevalent in the general population [1,2,3,4,5,6,7,8,9]. The illnesses underlying these symptoms are often unknown and treatment options are scarce, with a focus on nutrient supplements [10]. There is a potential causal association between these highly prevalent mental symptoms and low serum levels of B vitamins [11]. Pyridoxine (vitamin B6), folic acid (vitamin B9), and cobalamin (vitamin B12) are necessary cofactors for the synthesis of myelin and neurotransmitters [12,13,14,15,16,17]. A decrease in levels of these vitamins is associated with elevated homocysteine levels, risk of cognitive impairment [18,19,20,21,22,23], and depression [24,25]. The main cause of decreases in vitamin B12 serum levels is enteral malabsorption, primarily in the elderly due to gastric atrophy [26,27]. Mere food fortification is insufficient in this situation but high-dose oral or parenteral supplements increase vitamin B12 serum levels [28,29].

Diagnostic thresholds for vitamin B12 serum levels differ between countries and laboratories. For deficiency, serum levels below 148 pmol/L are most often used [14]. Subclinical vitamin B12 deficiency refers to serum levels above 148 pmol/L and the thresholds defining vitamin B12 adequacy are still debated. In people aged 60 years or older, the prevalence of vitamin B12 deficiency is about 5–6% and the prevalence of subclinical vitamin B12 deficiency is 20–25% [30,31]. The role of vitamin B12 “replacement” which applies to patients with unambiguous vitamin B12 deficiency as in pernicious anemia is undisputed. However, the role of vitamin B12 “supplementation” which applies to patients with subclinical or normal vitamin B12 serum levels is less clear. Considering the association of subclinical vitamin B12 serum levels with declines in cognitive function, depressive symptoms, and fatigue, vitamin B12 supplementation appears to be a promising treatment option. Therefore, physicians often prescribe vitamin B12 to improve mental outcomes across a broad range of vitamin B12 serum levels, including subclinical and normal levels. The current popularity of vitamin B12 supplementation is reflected by a recently published study from Canada, reporting that up to 60% of vitamin B12 prescriptions are given to patients aged 65 years or older with normal or undocumented vitamin B12 serum levels [32].

To date, several randomized trials (RCTs) and a few systematic reviews on B vitamins for mental outcomes have been published. Most of the original studies and systematic reviews have provided little support to continuing supplementation of B vitamins for improving mental outcomes; however, the applicability of the results in clinical practice is limited. This is because the former reviews included original studies, which included patients with advanced neurological disorders, leading to heterogeneous patient populations [33,34,35,36,37,38,39,40,41,42,43,44,45,46,47,48,49,50], or even focused exclusively on such patients [50]. Thereby, previous systematic reviews have provided little support for physicians, whose patients with mental ailments typically do not present detectable cerebral disease, and hence, vitamin B12 supplementation may still be an effective treatment option. Further, most systematic reviews have also included trials investigating folic acid alone, introducing heterogeneity at the treatment level, not considering the specific role of vitamin B12 [33,34,35,38,39,41,42,44,45,47,48,49,50,51].

The aim of this systematic review and meta-analysis of RCTs was to assess the effects of vitamin B12 supplementation—alone and in combination with folic acid—on cognitive function, depressive symptoms, and idiopathic fatigue, in patients without advanced neurological disorders.

## 2. Materials and Methods

This systematic review followed the guidelines of the “Preferred Reporting Items for Systematic Reviews and Meta-Analyses” (PRISMA) [52] and was registered at Prospero (CRD42019135823).

### 2.1. Data Sources and Search Strategy

An experienced librarian (S.K.) at Careum Bibliothek of the University of Zurich (Zurich, Switzerland) conducted a systematic literature search in June 2020. The following six databases were included: MEDLINE^®^ (via Ovid), Embase^®^, PsycINFO, Cochrane Library, and Scopus. The following search terms among others were used: “vitamin B12”, “cyanocobalamin”, “methylcobalamin”, “hydroxycobalamin”, “cognition”, “cognitive impairment”, “cognitive function”, “memory”, “depression”, “depressive”, and “fatigue”. A detailed description of all search strategies is provided in Appendix A. In addition, we searched the clinical trial registries ClinicalTrials.gov, International Clinical Trials Registry Platform (ICTRP), EU Clinical Trial Register, and the ISRCTN registry for ongoing or unpublished trials. No search limitations were applied in terms of publication date or language. Further, reference lists from reviews and meta-analyses were manually scanned to identify any other relevant studies.

### 2.2. Criteria for Study Inclusion/Exclusion

We included RCTs that investigated the therapeutic effects of vitamin B12 alone (B12 alone) and vitamin B12 in addition to vitamin B9 (folic acid) with or without vitamin B6 (B complex) on measures of cognitive function, depressive symptoms, and fatigue in patients with or without mild cognitive impairment (MCI). Eligible studies had to be double-blind and placebo-controlled with at least 20 participants aged 18 years or older per study group; administered a daily defined dose (DDD) of at least 100 mcg vitamin B12 either orally or intramuscularly. Studies were excluded if they were directed at patients with neuropathy following overt vitamin B12 deficiency (thus, studying vitamin B12 “replacement” and not “supplementation”), advanced neurological disorders (e.g., Alzheimer’s disease, dementia, or stroke), major depression or psychotic disorders (e.g., schizophrenia, psychosis); targeted highly specialized patient groups (e.g., with end-stage renal disease only, chronic intestinal diseases); if vitamin B12 was a component of a multivitamin supplement containing additional substances beyond vitamin B9 and B6; or if they had a high risk of bias after full text analysis (according to the Cochrane risk-of-bias tool, see quality assessment).

### 2.3. Study Selection, Quality Assessment, and Data Extraction

Title and abstract screening of all publications identified by the systematic literature search was conducted independently by two reviewers (S.M. and J.M.B.). Subsequently, a full text analysis for eligibility was performed. Disagreements were discussed and resolved by consensus or with third party arbitration (T.R.).

To assess the quality of the included RCTs, we used the revised Cochrane risk-of-bias tool for randomized trials (RoB 2) [53]. The study quality criteria included five domains: risk of bias arising from the randomization process, risk of bias due to deviations from the intended interventions (effect of assignment to intervention and of adhering to intervention), missing outcome data, risk of bias in measurement of the outcome, and risk of bias in selection of the reported result. Overall risk of bias judgement was evaluated as follows: low risk of bias if all domains were judged to be of low risk, some concerns if at least one domain was judged to raise some concerns but none to be at high risk, and high risk of bias if at least one domain was judged to be at high risk or multiple domains were judged to raise some concerns [53].

To assess the overall quality of evidence, we used the Grading of Recommendations Assessment, Development, and Evaluation (GRADE) tool [54]. The assessment was based on five key dimensions: risks of bias, inconsistency, indirectness, imprecision, and publication bias. The quality of the evidence was finally rated as high, moderate, low or very low.

Relevant data for each included trial were extracted into a Microsoft Excel (2016) spreadsheet (Microsoft Corporation, Redmond, WA, USA) by the first reviewer (J.M.B.), which included sample size, standard baseline patient characteristics (age, sex, and blood biochemical levels), doses of vitamin B12/B9/B6 and their administration frequency and administration route, treatment and follow-up duration, and outcomes. Data that were presented in figures only were made numeric using the WebPlotDigitizer [55]. All entered data were subsequently double-checked based on the original trials by a second reviewer (S.M.). If a trial only reported the results of subgroups, an inquiry was sent to the corresponding author to request data that were relevant for our systematic review analysis.

### 2.4. Outcomes

Primary endpoints were between-group differences on validated clinical tests measuring cognitive function, depression, or fatigue.

### 2.5. Data Synthesis and Analysis

Sixty-two clinical tests were used to measure one of the three different outcome domains, and many of them were only assessed in one or two trials. Cognitive function tests often targeted highly specific functions, and thus, we aggregated them into four outcome subdomains (cognitive executive, cognitive memory, cognitive global and cognitive speed) to maintain adequate precision in the analysis. Tests assessing depression and fatigue were each categorized into one domain (depressive symptoms and idiopathic fatigue, respectively). A short description of each clinical test can be found in Appendix A.

All analyses were conducted with R4.0.3 (R Foundation for Statistical Computing, Vienna, Austria) [56].

For the meta-analysis, we gathered and standardized data for the primary endpoints and calculated effect sizes (ES) as standardized mean differences (Hedges’ g) with respective confidence intervals (CI) in order to compare vitamin B12 interventions to placebos. Computations were carried out with the package esc [57]. Due to the presence of multiple outcome measures per study, we required meta-analysis models beyond the standard random-effects model, as we could not assume that the outcomes within each study were independent and estimated the same outcome. We, therefore, used three-level random-effects (3LRE) models with random effects at the single study level and of single outcome measures (the individual clinical tests) fitted with the package metafor [58]. For each outcome, we performed two separate meta-analyses for each of the two different intervention types (B12 alone and B complex) whenever more than one study was available and one meta-analysis for the pooled effect of both treatments. To assess differences between intervention types, subgroup analyses were carried out for outcomes encompassing at least 10 studies in mixed-effect models, using intervention type as a fixed-effect moderator, in addition to the 3LRE models as described in [59]. Statistical significance of intervention type was assessed by means of the t-test. The results of meta-analyses for the individual outcomes were summarized in forest plots showing the estimated summary effects of B12 alone, the B complex, and the pooled effect of B12 treatments. Between-study heterogeneity was reported in terms of the study-level random effect variance estimate tau-squared (τ2) and the I-squared (I2) statistic, computed for 3LRE models using the approach described in [60]. In addition, Cochran’s Q test was used to assess significance of study-level heterogeneity.

Meta-regression models were used to explore associations of treatment effects with potential moderators included as fixed effects within the 3LRE model framework. Moderators included were baseline vitamin B12 serum levels, vitamin intervention (B12 alone or B complex), duration of treatment, average bioavailable daily dose (assuming oral bioavailability of 1% [28,61,62] and intramuscular bioavailability of 15% [63]), the interaction between duration of treatment and average bioavailable daily dose, and population characteristics (only patients without MCI (=no MCI), only patients with MCI (=MCI), or mixed population samples (=no MCI to MCI)). We applied such meta-regression models when at least 10 studies for an outcome subdomain were available, following the recommendations by the Cochrane handbook for systematic reviews of interventions [64].

To assess if our results were dependent on the choice of model and parameters, we compared the aforementioned 3LRE model to a model based on robust variance estimation (RVE) with a small sample size adjustment, using the package robumeta [65,66].

To assess the possibility of publication bias, we used funnel plots and Egger’s test [67].

To accommodate for multiple hypothesis testing, we set the *p*-value threshold for statistical significance at 0.005 [68].

## 3. Results

### 3.1. Study Selection

As shown in Figure 1, 10,250 studies were retrieved by a systematic search and this figure was reduced to 7016 studies after deduplication. Based on relevant titles and abstracts, 53 studies were assessed in detail, applying inclusion and exclusion criteria as well as criteria for the methodological quality.

Two potentially eligible studies only reported the results of subgroup analyses [69,70]. Unfortunately, the corresponding authors were unable to provide us with the required data of the intervention and control groups (e.g., number of patients in their subgroups low and high tHcy and tertile 1–3, respectively) to calculate the ES and CI for the corresponding outcomes. Therefore, we had to exclude these two studies from our systematic review. Further, we had to exclude one pilot trial due to the high risk of bias after quality assessment [71].

Finally, 16 studies were eligible for our systematic review. The reasons for exclusion are depicted in Figure 1 and in Appendix A.

### 3.2. Overview and Characteristics of Included Studies

Of the 16 included studies, three investigated B12 alone versus a placebo [72,73,74], twelve studied the B complex versus a placebo [75,76,77,78,79,80,81,82,83,84,85,86], and one study investigated B12 alone versus the B complex versus a placebo [87]. Therefore, the latter study contributed separately to the B12 alone set as well as to the B12 complex set and counted as two studies for further analysis. Further, three [75,84,85], two [76,82], and two [77,78] trials were each based on the same study population; however, they contributed to different outcome domains or analyzed different subgroups.

Table 1 shows the baseline characteristics of the included studies, divided into B12 alone and B complex groups. The publication years ranged from 1996 to 2019, the mean age of the patients ranged from 66 to 82 years, the mean treatment duration ranged from 4 to 117 weeks, and the mean follow-up periods ranged from 8 to 117 weeks. Means or medians of baseline vitamin B12 serum levels ranged from 186 to 385 pmol/L and from 25.2 to 29.2 for the Mini-Mental State Examination (MMSE) scores. Further details about the baseline and study characteristics of the included studies can be found in Appendix A. Fifteen trials administered vitamin B12 orally with dosage ranges between 0.1 mg to 1 mg daily, and two intramuscularly with a dose of 1 mg once or twice weekly.

### 3.3. Quality Assessment of Included Studies and Publication Bias

Appendix A shows the risk of bias of all included RCTs. Twelve trials were found to have a low risk in all domains [72,73,74,75,76,77,80,82,83,84,85,86], five studies were judged to raise some concerns in one or two domains [78,79,81,87], and one study was at overall high risk (which was excluded before analysis) [71].

Results from the GRADE assessment are presented in Appendix A. The four cognitive function outcome subdomains, as well as the depression domain were rated as moderate quality.

We did not observe any asymmetries in the funnel plots for any cognitive function outcome subdomain nor for the depression domain (Appendix A).

### 3.4. Effects on Cognitive Function

In total, twelve studies reported effects on cognitive function, including four for B12 alone [72,73,74,87] and eight for the B complex [77,79,80,82,83,84,86,87]. Overall, we found no evidence for an effect of B12 alone and B complex supplementation on any subdomain of cognitive function outcomes. Even upon separate analyses of B12 alone or the B complex, there was no evidence of an effect (Table 2, Figure 2; Figure 3, Appendix A).

Meta-regression was performed for the cognitive subdomains measuring executive functions and memory (Appendix A). Meta-regression showed no significant associations of treatment effects with any of the potential predictors (baseline vitamin B12 serum levels, vitamin intervention, duration of treatment, average bioavailable daily dose, the interaction between duration of treatment and average bioavailable daily dose, or population characteristics). Since four studies did not contain baseline vitamin B12 serum levels, they were dropped from the full meta-regression model [78,81,83,86]. We explored the data further and included the four studies by removing the factor of baseline B12 serum level from the model, but the associations of the remaining factors with treatment effects were essentially unchanged (Appendix A).

For the outcome subdomains: cognitive speed and cognitive global, not enough studies (*n* = 8 and *n* = 7, respectively) were available for applying meta-regression models.

### 3.5. Effects on Depressive Symptoms

Effects on depressive symptoms were reported by six studies including one for B12 alone [73] and five for the B complex [75,76,78,85]. We found no overall effect of vitamin supplementation on measures of depression as well as not for B complex separately (Table 2, Figure 4).

A meta-analysis estimate for B12 alone and a meta-regression model were not possible due to insufficient data.

### 3.6. Effects on Idiopathic Fatigue

Only one study investigating the B complex reported effects on idiopathic fatigue [81]. Therefore, neither a meta-analysis estimate nor a meta-regression model were possible. Interestingly, the authors reported a statistically significant effect of vitamin B12 compared to placebo on the vitality subscale of the SF-36 survey only at the end of the follow-up period (8 weeks). Moreover, there was already a statistically significant difference at baseline between the two groups with higher starting scores (=better) in the placebo group.

### 3.7. Sensitivity Analyses

Due to the difficulty of modeling the potentially correlated outcome measures from each study, in addition to the 3LRE model, we considered the RVE model to explore sensitivity to model and parameter choice of our results. We fitted each of the two models using a variety of correlation parameters to the outcomes of cognitive and depression tests. We found that the RVE model was completely insensitive to correlation parameter choice and had similar point estimates to the 3LRE model but wider confidence intervals.

## 4. Discussion

### 4.1. Main Findings

In this study, we included 16 RCTs with low risks of bias, testing vitamin B12 supplementation in patients without advanced neurological disorders. The largest part of the available evidence concerned the effectiveness of vitamin B12 supplementation in subdomains of cognitive function. The meta-analysis found no effect of vitamin B12 supplementation on any of these subdomains. For depressive symptoms, the meta-analysis also showed no treatment effects. Regarding fatigue, the paucity of available high-quality evidence precludes robust conclusions at this time.

### 4.2. Relation to Previously Published Systematic Reviews

Previously published systematic reviews investigating the effects of B12 alone or B complex supplementation on cognitive function included studies performed in patients with advanced neurological disorders [33,34,35,36,37,38,39,40,41,42,43,44,45,46,47,48,49,50]. This complicated the real world applicability of the results because it was unclear as to which patients they applied, especially when considering that the majority of patients with mental aliments do not suffer from advanced neurological disorders. Thus, our study now contributes information that applies to a large proportion of patients. Furthermore, almost three quarters of the previous reviews also included trials that exclusively investigated folic acid or vitamin B6, introducing heterogeneity not only at the patient level but also at the treatment level.

All previous reviews, except two, found no evidence in favor of B vitamin supplementation and were, thus, in line with our own results. A very recently published meta-analysis by Suh et al. [48] reported beneficial effects for global cognition and episodic memory, which is in particularly stark contrast to our meta-analysis. Additionally, from including heterogeneous patient populations, the meta-analysis by Suh et al. also included studies with heterogeneous interventions. Most importantly, however, results based heavily on two studies, from which one was non-randomized [88] and the other was not placebo-controlled [89], introduce a high risk of bias. Another meta-analysis published in 2013 also found moderate beneficial effects on memory [42] but rests on only pooling results from two studies.

Only one previous systematic review exists on the effects of B vitamins on depression [51]. This systematic review also found no evidence in favor of B vitamin supplementation, but many trials included only patients with major depressive episodes, hampering the applicability of results because of heterogeneity at the patient level. In our study we excluded trials targeted at patients with major depression, reducing the heterogeneity and increasing the applicability to predominantly less severely depressed patients.

### 4.3. Strengths and Limitations

To the best of our knowledge, this is the most extensive systematic review and meta-analysis, including only high-quality RCTs that investigated the effect of vitamin B12 in the absence of advanced neurological disorders on cognitive function, depressive symptoms, and fatigue. The strengths of this systematic review are its applicability to large segments of the general population. In this context, it is important to consider that significant amounts of vitamin B12 prescriptions are currently given to patients with normal vitamin B12 serum levels [32]. Second, no other systematic review and meta-analysis used meta-regression analysis to explore potential determinants of the effectiveness of B12 supplementation, such as treatment duration and dose. Third, this is the first systematic review to assess vitamin B12 supplementation for idiopathic fatigue and our comprehensive literature review showed that very little high-quality evidence is available on this topic. While this scarcity of evidence hindered the meta-analysis, it also strongly contrasted with popular beliefs and claims from marketing campaigns suggesting the effectiveness of vitamin B12 supplementation for fatigue. Ultimately, we used the GRADE tool to assess the overall quality of the evidence.

One limitation of this study is the potentially large time difference between the onset of vitamin B12 supplementation and the appearance of measurable benefits. Currently, available RCTs, however, cover treatment durations of up to two years and, therefore, the question of the effectiveness of long-term vitamin B12 supplementation remains unanswered due to the lack of experimental evidence. Another limitation of this study originates from the remaining heterogeneity of included trials. While our inclusion criteria reduced the clinical baseline heterogeneity of patient populations, we nevertheless encountered trials reporting multiple outcomes for which one cannot assume independence, and, therefore, standard random-effects models are inadequate. Statistical modeling in this domain is still under development and best practices have not yet been established. Therefore, we implemented two different models to account for dependence between outcomes within studies and explored model dependency of effect estimates. This sensitivity analysis showed that our conclusions were robust for model choice. As there were no outliers among the included trials, the statistical heterogeneity was very small, which adds to the credibility of our pooled effect estimates [90]. Moreover, even if we excluded studies investigating vitamin B12 replacement in pernicious anemia or other forms of clinically relevant vitamin B12 deficiency, some individuals in the included trials might still have suffered from such conditions because these are sometimes difficult to diagnose and only a minority of included trials formally excluded such patients. Likewise, vitamin B12 levels alone are of limited predictive value for vitamin B12 deficiency and comprehensive baseline estimates more precisely determine vitamin B12 deficiency (including homocysteine, methylmalonic acid and holotranscobalamin) were only used by a minority of the RCTs included in this study. Our data are, therefore, potentially contaminated by individuals not receiving vitamin B12 “supplementation” but rather “replacement” due to them indeed being vitamin B12 deficient. Such a contamination, however, would have biased our results in the direction of the effectiveness of vitamin B12 supplementation because in deficient individuals, beneficial effects of vitamin B12 are more likely to occur. Therefore, our finding of the lack of effectiveness remains unchallenged by this particular source of bias. Lastly, it must be highlighted that this systematic review was exclusively concerned with vitamin B12 supplementation in individuals without overt vitamin B12 deficiency and also in general primary care populations without specific risk factors for vitamin B12 malnutrition, such as meat-free diets. The role of vitamin B12 replacement in vitamin B12 deficiency, as well as vitamin B12 supplementation in the risk of malnutrition, such as in vegan diets is not addressed in this systematic review.

### 4.4. Controversies Raised by This Study

Vitamin B12 supplementation continues to be a popular treatment even in patients without overt vitamin B12 deficiency [32]. Many reasons may contribute to this popularity of vitamin B12 supplementation, and in addition to expecting some specific—albeit weak—effects, vitamins are popular for use as placebo interventions in cases where no other treatment options are available [91]. Such liberal use of vitamin B12 supplementation, however, might encounter criticism in the light of recent observational studies linking vitamin B12 supplementation or serum levels with hip fractures [92], lung cancer [93,94], and all-cause mortality in the general population [95].

### 4.5. Future Research

In the context of cognitive functioning and depressive symptoms, this systematic review suggests that there is a low likelihood that future studies of similar design will overturn the aggregated evidence. Since the longest treatment duration investigated was two years, the most significant knowledge gap seems to concern long-term vitamin B12 supplementation. Interestingly, for fatigue, there is currently only one RCT of high quality and there is, thus, room for future studies to fill this knowledge gap. Moreover, the role of vitamin B12 supplementation could still be better understood in increasingly popular meat-free diets.

## 5. Conclusions

This systematic review, focusing on vitamin B12 supplementation for elderly patients with normal or subclinical vitamin B12 serum levels and without advanced neurological disorders, found high-quality evidence for the absence of treatment effects on cognitive functioning and depressive symptoms. For idiopathic fatigue, this literature review detected a paucity of evidence. Taken together, the findings from this systematic review discourage vitamin B12 supplementation for cognitive function and depressive symptoms.

## Figures and Tables

**Figure 1 nutrients-13-00923-f001:**
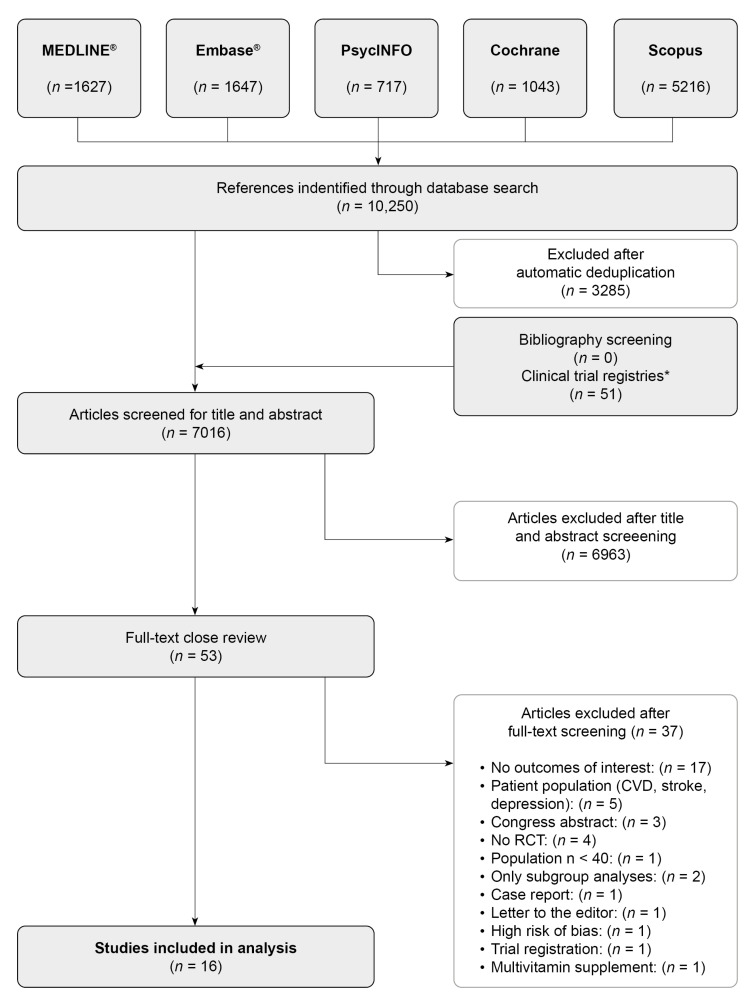
Preferred Reporting Items for Systematic Reviews and Meta-Analyses (PRISMA) flow diagram. * ClinicalTrials.gov, International Clinical Trials Registry Platform (ICTRP), EU Clinical Trial Register, and ISRCTN registry.

**Figure 2 nutrients-13-00923-f002:**
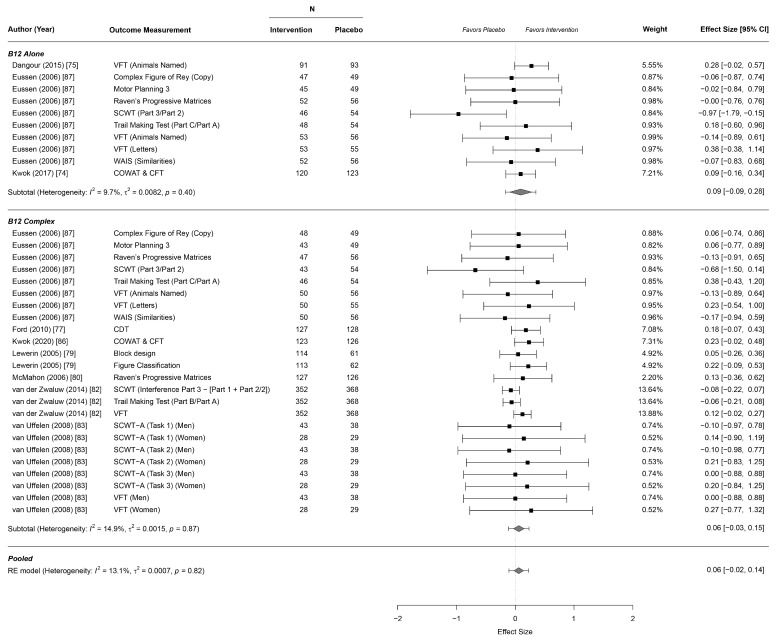
Forest plot for effects on cognitive executive function. CDT: Clock Drawing Test; CFT: Category Fluency Test; COWAT: Controlled Oral Word Association Test; SCWT: Stroop Color–Word Test; SCWT-A: Stroop Color–Word Test Abridged; VFT: Verbal Fluency Test; WAIS: Similarities Wechsler Adult Intelligence Scale.

**Figure 3 nutrients-13-00923-f003:**
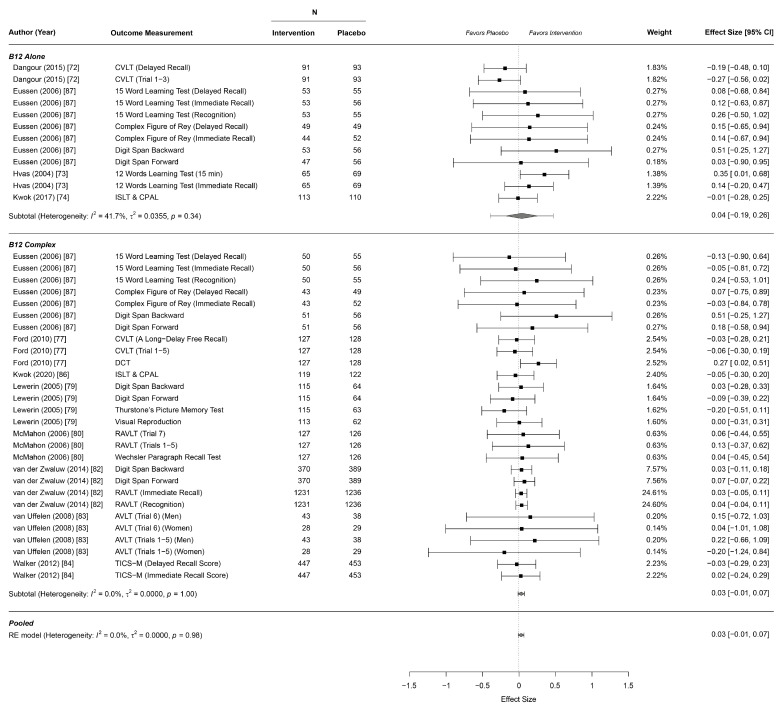
Forest plot for effects on cognitive memory function. AVLT: Auditory Verbal Learning Test; CPAL: Continuous Paired Associates Learning; CVLT: California Verbal Learning Test; DCT: Digit Cancellation Test; ISLT: International Shopping List Test; RAVLT: Rey Auditory Verbal Learning Test; TICS-M: Telephone Interview for Cognitive Status Modified.

**Figure 4 nutrients-13-00923-f004:**
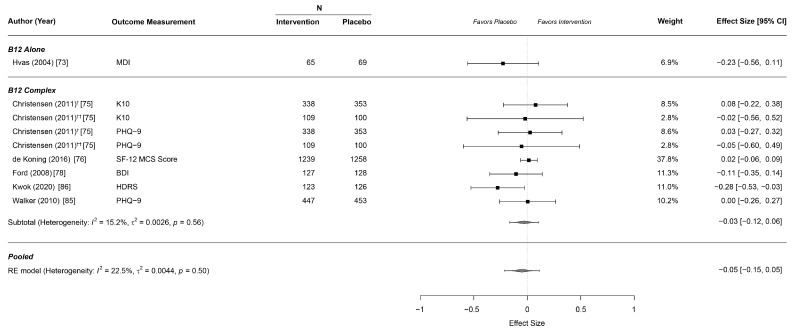
Forest plot for effects on depressive symptoms. † with antidepressant; †† without antidepressant. BDI: Beck Depression Inventory; HDRS: Hamilton Depression Rating Scale; K10: Kessler Psychological Distress Scale; MDI: Major Depression Inventory; MCS: PHQ-9: Patient Health Questionnaire 9.

**Table 1 nutrients-13-00923-t001:** Patient characteristics at baseline of included studies.

Author, Year	Participants, n	Female, n (%)	Mean Age (SD), Years	Population Characteristics	Vitamin B12 Serum Level (SD), pmol/L (I: Intervention P: Placebo)	Vitamin B12 (DDD) Administration, mcg	Vitamin B9 Administration, mcg	Vitamin B6 Administration, mg	Intake Frequency, Route of Administration	Treatment Duration, Follow-Up Duration, Weeks	Outcome Domain
B12 alone											
Dangour, 2015 [72]	201	107 (53.2)	80.0 (3.7)	No MCI	I: 222.9 (197.4–268.9) ^a^ P: 228.0 (194.7–271.0) ^a^	1000 (1000)	n.a.	n.a.	daily, oral	52, 52	Cognitive
Eussen, 2006 [87]	195	149 (76.4)	82.3 (5.0)	No MCI to MCI	I: 186.0 (56.0) P: 188.0 (56.0)	1000 (1000)	n.a.	n.a.	daily, oral	24, 24	Cognitive
Hvas, 2004 [73]	140	98 (7.0)	n.a. (n.a.)	No MCI to MCI	I: 278.0 (143.0–1348.0) ^b^ P: 254.0 (137.0–724.0) ^b^	1000 (143)	n.a.	n.a.	weekly, intramuscular	4, 12	Cognitive, depression
Kwok, 2017 [74]	271	113 (41.7)	75.3 (4.2)	No MCI to MCI	I: 227.5 (40.0) P: 235.9 (37.9)	2 × 500 (1000)	n.a.	n.a.	daily, oral	117.5, 117.5	Cognitive
B complex											
Christensen, 2011 [75]	900	542 (60.2)	65.9 (4.4)	No MCI	I: 305.0 (151.0) P: 285.0 (106.0) I2 ^c^: n.r. P2 ^c^: n.r.	100 (100)	400	n.a.	daily, oral	104, 104	Depression
de Koning, 2016 [76]	2919	1459 (50.0)	74.1 (n.a.)	No MCI	I: 267.0 (213.0–341.0) ^a^ P: 266.0 (204.0–343.0) ^a^	500 (500)	400	n.a.	daily, oral	104, 104	Depression
Eussen, 2006 [87]	195	100 (51.3)	82.5 (6.0)	No MCI to MCI	I: 199.0 (50.0) P: 188.0 (56.0)	1000 (1000)	400	n.a.	daily, oral	24, 24	Cognitive
Ford, 2008 [78]	299	0 (0.0)	79.0 (2.7)	No MCI	I: n.r. P: n.r.	400 (400)	2000	25	daily, oral	104, 104	Depression
Ford, 2010 [77]	299	0 (0.0)	79.0 (2.8)	No MCI	I: 256.12 (121.86) P: 253.02 (115.35)	400 (400)	2000	25	daily, oral	104, 104	Cognitive
Kwok, 2020 [86]	279	113 (40.5)	77.5 (n.a.)	MCI	I: n.r. P: n.r.	500 (500)	400	n.a.	daily, oral	52, 104	Cognitive, depression
Lewerin, 2005 [79]	209	117 (55.9)	75.7 (4.7)	No MCI	I: 305.0 (130.0) P: 359.0 (198.0)	500 (500)	800	3	daily, oral	16.92, 16	Cognitive
McMahon, 2006 [80]	276	112 (44.3)	73.5 (5.8)	No MCI	I: 380.0 (136.0) P: 385.0 (138.0)	500 (500)	1000	10	daily, oral	104, 104	Cognitive
Schlichtiger, 1996 [81]	213	150 (70.4)	73.3 (5.9)	No MCI	I: n.r. P: n.r.	1000 (286)	1100	5	2/weekly, intramuscular	4, 8	Fatigue
van der Zwaluw, 2014 [82]	2919	1459 (50.0)	74.1 (6.5)	No MCI	I1: 267.0 (231.0–341.0) ^a^ P1: 266.0 (204.0–343.0) ^a^ I2 ^d^: 257.0 (200.0–326.0) ^a^ P2 ^d^: 263.0 (200.0–345.0) ^a^	500 (500)	400	n.a.	daily, oral	104, 104	Cognitive
van Uffelen, 2008 [83]	179	67 (37.4)	75.17 (n.a.)	MCI	I: n.r. P: n.r.	400 (500)	5000	50	daily, oral	52, 52	Cognitive
Walker, 2010 [85]	900	542 (60.2)	66.0 (4.3)	No MCI	I: 305.32 (151.05) P: 285.27 (105.77)	100 (100)	400	n.a.	daily, oral	104, 104	Depression
Walker, 2012 [84]	900	542 (60.2)	66.0 (4.3)	No MCI	I: 305.32 (151.05) P: 285.27 (105.77)	100 (100)	400	n.a.	daily, oral	104, 104	Cognitive

DDD: daily defined dose; DS: depressive symptoms; MCI: mild cognitive impairment. ^a^ median [IQR]; ^b^ median [range]; ^c^ with antidepressant; ^d^ extensive cognitive tests subsample.

**Table 2 nutrients-13-00923-t002:** Effects on cognitive function and depression.

Outcome (sub)Domain	Effect Size	95% CI LB	95% CI UB	*I* ^2^	τ^2^	*p-*Value
**Vitamin B12 Alone**						
Cognitive executive	0.09	−0.09	0.28	9.7%	0.0082	0.40
Cognitive memory	0.04	−0.19	0.26	41.7%	0.0355	0.34
Cognitive global	0.02	−0.17	0.21	6.3%	0.0018	0.56
Cognitive speed	−0.10	−0.23	0.04	0%	0	0.84
Depression	n.a.	n.a.	n.a.	n.a.	n.a.	n.a.
**Vitamin B Complex**						
Cognitive executive	0.06	−0.03	0.15	14.9	0.0015	0.87
Cognitive memory	0.03	−0.01	0.07	0.0	0	1.00
Cognitive global	0.07	0.00	0.13	0.0	0	0.93
Cognitive speed	−0.08	−0.25	0.10	43.4	0.0149	0.68
Depression	−0.03	−0.12	0.06	15.2	0.0026	0.56
**Pooled (Vitamin B12 and Vitamin B Complex)**						
Cognitive executive	0.06	−0.021	0.141	13.1	0.0007	0.82
Cognitive memory	0.028	−0.011	0.067	0.0	0	0.98
Cognitive global	0.061	−0.001	0.123	0.0	0	0.95
Cognitive speed	−0.081	−0.175	0.013	13.9	0.0037	0.88
Depression	−0.049	−0.146	0.047	22.5	0.0044	0.50

LB: lower bound; UB: upper bound; n.a.: not applicable.

## Data Availability

The data presented in this study are openly available in Mendeley Data at [10.17632/t44jhwfdj5.1].

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
