# Peer review of "Effects of Vitamin B12 Supplementation on Cognitive Function, Depressive Symptoms, and Fatigue: A Systematic Review, Meta-Analysis, and Meta-Regression"

_nutrients, 2021, doi:10.3390/nu13030923_

Round 1
Reviewer 1 Report
This is a well written and well executed meta-analytic review of RCTs testing the effects of vitamin B12 and/or B complex supplementation on changes in cognition, depression and fatigue. I have a few comments regarding improving the paper.
- Although I appreciate what the authors were trying to do in terms of excluding people with conditions, the resulting selection of studies seems to have favored participants with normal B12 levels. Why would supplementation of B12 improve cognitive performance (or depression or fatigue) in people with normal B12 levels? Could baseline B12 levels or average B12 levels in the study sample account for heterogeneity in outcomes? More needs to be acknowledged in the title, abstract, and discussion that these RCTs mainly tested their interventions on people with normal B12 levels. Are effect sizes larger in samples with lower baseline B12?
-
In section 3.6, it is written "Only one study investigating B complex reported effects on idiopathic fatigue [77]." Briefly mention what they found, especially since this study is difficulty to find (possibly only published in German), thus people may not be able to read the findings from the article. Did they find effects favoring supplementation or placebo? just a brief mention would be helpful.
- For the future research section in the discussion, you should mention the importance of future B12 research in vegans and vegetarians, and whether supplementation of B12 in these populations has the potential to improve cognitive processes, depression, and fatigue. Veganism and vegetarianism are becoming increasingly popular, with the result that vitamin B12 deficiencies may also follow such trends. To what extent does your work speak to this issue, and how could future work speak to this issue? This is relevant to point 1 above because vegans and vegetarians have below normal B12 levels (if not already supplementing or eating fortified foods), and experimental research shows a vegan diet reduces B12 levels in just four weeks:
https://pubmed.ncbi.nlm.nih.gov/31752105/
so if you are only meta-analysing data from people with near normal B12 levels, your data might not speak to what happens if you supplement in people with below normal B12 levels like vegans or vegetarians. Other relevant work here is: https://www.sciencedirect.com/science/article/pii/S0261561418300712
Reviewer 2 Report
This manuscript presents the results of a timely systematic literature review, meta-analysis and meta-regression that examined the effectiveness of Vitamin B 12 interventions for improving cognitive function, depression and fatigue. The paper is well-written and comprehensive account of the current state of the evidence on this topic, but there are some minor issues that need to be addressed:
Introduction:
- In the first sentence, the authors should clarify that they are referring to an aging or older population rather than the general population as currently stated based on the references that they have provided for this statement
- What is the reason for low Vitamin B serum levels in this population i.e, is this insufficiency due to poor dietary intake in older adults?
- It would be useful to define what is meant by idiopathic fatigue
Method
- Was dietary status of participants included as a moderator or controlled for in the included studies? Special diets such as vegetarian or vegan may result in low Vitamin B12 levels
- What was the rationale for including studies with participants aged 18 or over? The introduction suggests that the focus of the review would be on older adults given the high supplementation rates in this group. Regardless, it would be interesting to determine the effects of age on the effectiveness of the intervention. What was the age range of the studies overall? Was age a moderator of treatment effects?
- Table 1 should be presented in landscape format to improve readability
- Maybe I missed it, but did the authors assess for any differences between patients with subclinical and normal vitamin B12 serum levels?
Discussion
- The authors should acknowledge that the review was also limited to studies administering Vitamin B supplements rather than foods or food products (e.g. functional foods) that had been specifically formulated or fortified to provide high dose Vitamin B
- Did any of the included studies control for or measure diet quality of the participants?
- Line 367-9: It is unclear whether these studies suggest that Vitamin B12 causes such health conditions and if so, what is the level of evidence? These studies appear to be observational in nature rather than for example, reported adverse events occurring as part of a Vitamin B 12 human clinical trial
